# Assessment of Tissue Distribution and Metabolism of MP1, a Novel Pyrrolomycin, in Mice Using a Validated LC-MS/MS Method

**DOI:** 10.3390/molecules25245898

**Published:** 2020-12-13

**Authors:** Wafaa N. Aldhafiri, Yashpal S. Chhonker, Yuning Zhang, Don W. Coutler, Timothy R. McGuire, Rongshi Li, Daryl J. Murry

**Affiliations:** 1Department of Pharmacy Practice and Science, College of Pharmacy, University of Nebraska Medical Center, Omaha, NE 68198, USA; wafaa.aldhafiri@unmc.edu (W.N.A.); y.chhonker@unmc.edu (Y.S.C.); zhangyuning0315@hotmail.com (Y.Z.); guanxing95@yahoo.com (T.R.M.); 2Department of Pharmacy, Nanfang Hospital, Southern Medical University, Guangzhou 510515, China; 3Division of Pediatrics, College of Medicine, University of Nebraska Medical Center, Omaha, NE 68198, USA; dwcoulter@unmc.edu; 4Department of Pharmaceutical Sciences, College of Pharmacy, University of Nebraska Medical Center, Omaha, NE 68198, USA; rongshi.li@unmc.edu; 5Fred and Pamela Buffett Cancer Center, University of Nebraska Medical Center, Omaha, NE 68198, USA

**Keywords:** pyrrolomycin, in-vitro metabolism, LC-MS/MS, biodistribution

## Abstract

MP1 is a novel marinopyrrole analogue with activity in MYCN amplified neuroblastoma cell lines. A rapid, selective, and sensitive liquid chromatography coupled with tandem mass spectrometry (LC-MS/MS) method was developed and validated for quantitation of MP1 in mouse plasma. Analyte separation was achieved using a Waters Acquity UPLC^®^BEH C18 column (1.7 µm, 100 × 2.1 mm). Mobile phase consisted of 0.1% acetic acid in water (10%) and methanol (90%) at a total flow rate of 0.25 mL/min. The mass spectrometer was operated at unit resolution in the multiple reaction monitoring (MRM) mode, using precursor ion > product ion transitions of 324.10 > 168.30 *m*/*z* for MP1 and 411.95 > 224.15 *m*/*z* for PL-3. The MS/MS response was linear over the concentration range from 0.2–500 ng/mL for MP1, correlation coefficient (r^2^) of 0.988. Precision (% RSD) and accuracy (% bias) were within the acceptable limits as per FDA guidelines. MP1 was stable under storage and laboratory handling conditions. The validated method was successfully applied to assess the solubility, in-vitro metabolism, plasma protein binding, and bio-distribution studies of MP1.

## 1. Introduction

Neuroblastoma (NB) is a rare embryological malignancy derived from neural crest cells. NB is mainly diagnosed before the age of 5 years, with a median age of diagnosis of 18 months. NB is the second most common extracranial malignant tumor in childhood and accounts for about 15% of pediatric cancer deaths [1,2,3,4]. Approximately 60% of children diagnosed with NB have metastatic disease at diagnosis and undergo resistance conventional therapy, resulting in poor survival [1,3].

Several genetic alterations have been observed in NB cells, including amplification of MYCN (V-Myc Avian Myelocytomatosis Viral Oncogene Neuroblastoma Derived Homolog). MYCN amplification is associated with a poor survival rate of only 14–34%, even in patients with favorable outcome profiles [5,6]. Amplification of MYCN supports oxidative glycolysis, also known as the Warburg effect, by activating the transcription of several glycolytic genes to meet the NB cells elevated need for glucose. Inhibition of MYCN pathways may lead to autophagy, quiescence and cell death and be a novel target for drug therapy of NB [4,7]. Despite advances in therapy, treatment of pediatric NB with MYCN amplification remains challenging [8]. There is a critical need to identify agents that target the MYCN pathway and improve outcomes in pediatric NB.

Marinopyrroles are novel alkaloids isolated from a marine Streptomyces (actinomycete strain CNQ-418) with an uncommon 1,3′-bipyrrole pharmacophore [9]. They were first reported as promising antibiotic agents. Recent studies have identified these compounds to have anti-myeloid cell leukemia 1 (Mcl-1) activity, suggesting they have anticancer activity [7]. Mcl-1 is an anti-apoptotic protein, highly expressed in a variety of human cancers and a validated drug target for cancer treatment [9,10]. Although marinopyrroles represent a novel class of anticancer compounds, their physicochemical and drug-like properties are not ideal for drug development. The majority of marinopyrroles have poor solubility, limiting enthusiasm for further development to an orally active drug. Marinopyrroles with improved physicochemical and drug-like properties have recently been described [7]. MP1, a novel marinopyrroles/pyrrolomycins derivative, has been shown to have activity in a resistant NB cell line overexpressing MYCN by inhibiting MYCN and MCL-1, while stimulating autophagy and inhibition of oxidative phosphorylation (OXPHOS) metabolism [7].

In order to characterize in-vitro properties and assess the pharmacokinetics and biodistribution of MP1, we developed a rapid and sensitive bioanalytical method for the quantification of MP1 with a short run time (~6 min). The results of these studies provide the needed information to guide MP1 dosing in animal models of NB. Further, these findings in correlation with biological response will be useful for further evaluation of the therapeutic use of MP1.

## 2. Results

### 2.1. Chromatographic and Mass Spectrometric Conditions Optimization

To optimize MS/MS conditions for MP1 and PL3 internal standard (IS) detection, electrospray ionization (ESI) and atmospheric pressure chemical ionization (APCI) conditions were tested. Signal intensities of MP1 and the IS were found to be the highest using ESI in the negative ionization electrospray ionization mode (ESI−) (data not shown). MS/MS conditions were optimized using 1 µg/mL of MP1. De-protonated molecules [M–H]^−^ of MP1 were seen in abundance, having a precursor ion > product ion combinations of 324.10 > 168.30 *m*/*z* and 411.95 > 224.15 *m*/*z* for PL3, respectively (Figure 1).

In order to attain short retention times and maximal peak resolution, various chromatographic conditions were investigated including: mobile phase (acetonitrile, MeOH, and water), additives (acetic acid, ammonium acetate, formic acid, and ammonium format), gradient parameters, and analytical columns (C8, C18, and C18 PFP). Final separation conditions were achieved on an Acquity UPLC^®^BEH C18 column, 1.7 µm, 100 × 2.1 mm, Waters, Inc. Milford, MA, USA) column equipped with an Acquity UPLC C18 guard column (Waters, Inc. Milford). Final mobile phase consisted of 0.1% acetic acid in water (mobile phase A; 10%), and MeOH (mobile phase B; 90%), utilizing an isocratic elution at a total flow rate of 0.25 mL/min. The retention time of MP1 was 1.9 min and 4.5 min for PL3. The method incorporated an additional 2.1 min after the final analyte peak, to wash out any residue and assure complete evaluation of the compound, bringing the total run time to 6.0 min. No interference from endogenous components were found in the biological matrix at MP1 or the IS retention time (Figure 1).

To optimize sample preparation, protein precipitation, liquid–liquid extraction and solid-phase extraction (SPE) methods were evaluated. The final SPE method was selected to prepare plasma and tissue samples due to its high extraction recovery. By comparing the SPE rate with various organic washing and eluting solvents, MeOH was finally selected as a reliable eluting solvent for SPE because it had good recovery with no matrix effect [11,12]. The extraction method showed reproducible and consistent recovery at all the evaluated concentrations in plasma and tissue samples.

### 2.2. Assay Validation

#### 2.2.1. Selectivity and Sensitivity

To analyze the specificity of the method, blank extracted plasma and tissue homogenate was analyzed for potential interferences at the retention times of 1.9 min (MP1) and 4.5 min (IS). There was no significant interference or co-eluting peaks that were >20% of the analytes area at the Lower limit of quantification (LLOQ) level, and no co-eluting peaks >5% of the area of IS were observed, meeting the acceptance criteria and indicating the specificity of the method.

The LLOQ for MP1 was determined to be 0.2 ng/mL. The intra–run precision at the LLOQ plasma samples containing MP1 was 10.8%. The intra–run accuracy at the LLOQ plasma samples was 7.3%. The intra and inter-day variability results are tabulated in Table 1.

#### 2.2.2. Calibration Curve and Linearity

The MP1 calibration curve was linear over the range of 0.2 to 500 ng/mL (Figure 2). The correlation coefficient of determination (r^2^) was greater than 0.987.

#### 2.2.3. Carry-Over

In order to assess carry-over, blank samples (running buffer only) were analyzed immediately after running high quality control (HQC) samples. No significant peaks, ≥20% LLOQ, were observed. Thus, it was concluded that carry over was negligible in the current method (<5% of LLOQ).

#### 2.2.4. Accuracy and Precision 

Intra- and inter-day accuracy and precision for MP1 was assessed at four different concentrations in mouse plasma LLQC (0.2 ng/mL), LQC (0.6 ng/mL), MQC (100 ng/mL) and HQC (375 ng/mL) levels. The %RSD of precision values ranged from 1.4 to 13.6%, indicating acceptable assay precision. The accuracy of the quantitative analysis of the compounds was varied from −13.4–7.3%, within the acceptance limits at all concentration levels. The accuracy and precision results are listed in Table 1.

#### 2.2.5. Recovery and Matrix Effect 

Analyte recovery for MP1 and IS from spiked plasma samples was calculated using three control samples (LQC—0.6 ng/mL, MQC—100 ng/mL and HQC—375 ng/mL). The % extraction mean recovery for MP1 in LQC, MQC and HQC was 95.8 ± 2.6, 89.1 ±10.6 and 90.8 ± 3.4%, respectively. The mean recovery of all QC levels was >90%, whereas the mean recovery of IS was 87.3 ± 3.8%. The matrix effect was negligible and within an acceptable ±15%. (Table 2)

#### 2.2.6. Stability

Stability of MP1 was assessed by several means including 4 h bench-top at room temperature 21 °C, three freeze-thaw cycles, autosampler stability for 24 h at temperature of 4 °C, and long term stability for 12 months frozen −80 °C, MP1 showed no appreciable instability, as concentrations were within the accepted ±15% standard (Table 3).

#### 2.2.7. Dilution Integrity

The dilution integrity of the samples was ensured by diluting QC samples above the upper limit of quantification (ULOQ) with like matrix to bring to within quantitation range 0.2–500 ng/mL. The mean analyzed value of all diluted samples was within 15% of nominal concentration, with precision of the replicates ±15% coefficient of variation (data not shown).

#### 2.2.8. Solubility of MP1

The solubility of MP1 in phosphate buffer (pH 7.4) was approximately 50 µg/mL (Mean ± SD 51.19 ± 2.11 µg/mL).

### 2.3. In Vitro Studies

#### 2.3.1. Gastrointestinal Stability of MP1

The stability of MP1 was evaluated at pH 1.2, 6.8, and pH 7.4. MP1 was found to be stable with >99% parent remaining after 1 h in simulated gastric fluid (SGF), pH 1.2, and 2 h of incubation in simulated intestinal fluid, (SIF) pH 6.8, and in buffer at pH 7.4. (Figure 3)

#### 2.3.2. The Blood to Plasma Ratio (Kb/p)

The blood:plasma (Kb/p) ratio was determined for MP1 at a concentrations of 1 µM (325 ng/mL). The Kb/p ratio was 0.40, 0.68 and 0.63 after 0, 30- and 60-min incubations at 37 °C (Table 4). 

#### 2.3.3. Plasma Protein Binding (PPB) Study 

MP1 was highly bound (>95%) to mouse plasma proteins at both 1 µg/mL and 10 µg/mL concentrations. MP1 was stable in mouse plasma following a 5 h incubation (Table 5). There was no binding of MP1 to the dialysis membrane (data not shown).

#### 2.3.4. In-Vitro Metabolic Stability in Microsomes

In-vitro metabolic stability of MP1 was investigated using mouse, rat, and human liver microsomes. The time-dependent metabolic depletion of MP1 in the interspecies microsomes is shown in Figure 4a. The in vitro t_½_ of MP1 was 15.64 ± 0.46, 18.1 ± 3.73, 27.47 ± 2.68 min in mouse, rat, and human, respectively. The in-vitro half-life (t_½_), intrinsic clearance (CL_int_) and in-vitro hepatic clearance (CL_int,H_) of MP1 in mouse, rat, and human microsomes are summarized in Table 6.

#### 2.3.5. In-Vitro Metabolic Stability in Mouse S9 Fraction 

In-vitro metabolic stability of MP1 was investigated using mouse S9 fractions. The result of the metabolic stability study was expressed as the % parent remaining at different time points relative to the parent at 0 min (100% parent) (Figure 4b). The in-vitro t_1/2_ of MP1 was 4.9 ± 0.2 min and CL_int_ of 142.3 ± 5.6. CL_int_ was calculated using the MP1 depletion over time from initial substrate concentration of 1 µM (Table 6). Positive controls utilizing testosterone, 7-HC and diclofenac were within the acceptable in-house limits.

#### 2.3.6. Metabolite Identification

Mouse S9 fraction samples were run in a 35 min gradient method using liquid chromatography coupled with tandem mass spectrometry (LC-MS/MS) in positive and negative ESI mode. Each peak, retention time, and precursor ion peak [M–H]^−^ was recorded, and structural information was collected by fragmentation of all precursor ions. The mass of each predicted metabolite was used to generate post-acquisition extracted ion chromatograms (EICs).

The parent compound MP1 was detected as a de-protonated molecular ion [M–H]^−^ with a retention time of 25.39 min. Two metabolites were identified based on their LC/MS data and putative structure via possible biotransformation, precursor ion, and retention times (Figure 5), and tabulated in Table 7. These results identify glucuronidation as the major metabolic pathway for MP1.

### 2.4. In-Vivo Pharmacokinetic Studies

#### 2.4.1. Pharmacokinetic Study

The developed and validated LC-MS/MS method was successfully applied to MP1 pharmacokinetics and tissue distribution studies, following a single 15 mg/kg oral dose of MP1 to female BALB/c mice (Table 8). The blood concentration vs. time profile for the MP1 is shown in Figure 6.

Pharmacokinetic data was processed with Phoenix^®^ 8.2 (Certara Corporation, Mountain View, CA) software simulating through non-compartmental analysis. The mean non-compartmental pharmacokinetic parameters are shown in Table 9. Following oral administration, MP1 C_max_ was 4714.7 ± 2343.5 ng/mL at a mean T_max_ of 0.6 h. MP1 t_1/2_ was 9.2 ± 1.7 h and CL was 1.1 ± 0.3 L/h/kg.

#### 2.4.2. Tissue Distribution Study 

Concentrations of MP1 was detected in all studied tissues, with the concentration the highest in the liver at all-time points evaluated (Table 10). Concentrations of MP1 were detectable at 48 h in all tissues (Figure 7).

## 3. Discussion

We successfully developed and validated a rapid, selective, and sensitive LC-ESI-MS/MS method for the quantitation of MP1 in mouse plasma and tissues. The response was linear over the concentration range of 0.2–500 ng/m with a correlation coefficient (r^2^) greater than 0.987 for all calibration curves. The method was validated according to FDA guidelines [13], showing accuracy and precision within the acceptance limits set forth by the FDA. The assay was sensitive with LLOQ of 0.2 ng/mL utilizing 100 µL of plasma.

The optimized MS conditions utilized ESI− and resulted in lower background noise and acceptable sensitivity for the quantitation of MP1 [11,14]. LC separation was performed using reversed phase (C18), which is wildly used by researchers for the relatively high organic content for solute interaction and better stability at both extremes of the working pH range. Because we used a reversed-phase stationary we were able to employ a polar mobile system consists of an isocratic elution of 0.1% AA in water (10% mobile phase A) and MeOH (90% mobile phase B). The use of the polar mobile system resulted in acceptable separation with high peak efficiency, peak symmetry, and short total run time of 6 min, making the method time efficient [11,12,15,16].

Sample extraction and clean up was achieved using SPE. The developed SPE protocol was created while considering the drug physical-chemical properties and compatibility with the chosen LC method. The extraction method was reproducible with consistent high extraction recovery averaged at >90% of MP1 and >85% of IS, and negligible matrix effect [15,16,17]. 

The stability validation of MP1 was determined under storage and use conditions, which emulate the laboratory handling conditions. MP1 was stable under the tested conditions: 4 h bench-top at 21 °C, three freeze-thaw cycles, 24 h, autosampler stability at 4 °C, and 12 months frozen −80 °C. Knowing the stability of MP1 helps in selecting proper storage conditions and laboratory handling procedures to ensure high purity compound, hence more reproducible and reliable results [18]. Moreover, MP1 was found to be stable with >99% parent remaining in SGF, SIG, and at physiological pH. MP1 was highly bound (>95%) to mouse plasma proteins. The unbound drug concentration is considered to be pharmacologically and toxicologically active [19]. In addition, MP1 mouse blood to plasma ratio (Kb/p) was <1, indicating no significant partitioning of the compound into any of the specific component in the blood [20].

The metabolic stability of MP1 was evaluated in mouse, rat, and human liver microsomes, and mouse S9 fraction. MP1 intrinsic clearance was 88.72 ± 2.66, 80.01 ± 16.5 and 50.99 ± 5.25 (µL/min/mg protein) in mouse, rat, and human liver microsomes respectively. In-vitro mouse S9 metabolic investigations exhibited rapid metabolic degradation with two metabolites being detected, glucuronidation being the major metabolic pathway [21,22].

The method was successfully applied to pharmacokinetic studies of the oral administration of MP1. We found that MP1 was rapidly absorbed with a T_max_ of 0.6 h with a t_½_ of 9.2 h. MP1 was found in the systemic circulation up to 48 h. In addition, MP1 showed good penetration in all the tested tissues, with the highest concentration in the liver and the lowest accumulation in the brain.

We did not observe any phase I metabolites (oxidation, hydroxylation) in these initial studies. The disappearance rate of MP-1 in the S9 fraction studies was much faster compared to that observed in the MLM, RLM and HLM in the presence of UDPGA or NADPH, respectively (Figure 4). The glucuronide metabolite was found to be the dominant metabolite suggesting the direct glucuronidation of MP1 was the major clearance pathway. For confirmation, detailed metabolism studies and identification of individual CYP or UGT involvement in the metabolism of MP-1 will be performed.

### Study Limitations and Future Directions 

Following the pre-clinical studies investigating the biodistribution of MP1, we found that MP1 is rapidly distributed throughout the body. These studies provide the framework for future studies to design dosing regimens that achieve MP1 concentrations associated with activity in cell lines (i.e., the EC_50_). One of the limitations of this study is that our drug concentrations represent total drug concentrations. Typically, free drug concentrations correlate with therapeutic effects better than total drug concentrations. Future studies will assess tumor drug penetration and cellular uptake of MP1 to bridge the gap between drug dose, free drug concentrations, and tumor response.

## 4. Materials and Methods 

### 4.1. Chemicals and Materials 

MP1 (purity: ≥98%) and PL3 (purity: ≥99.9%), used as an internal standard (IS), were generously provided by Dr. Rongshi Li (UNMC). Methanol (MeOH), acetonitrile (ACN) and sodium acetate were purchased from Fisher Scientific (Fair Lawn, NJ, USA). All solvents were HPLC grade or higher. Water was purified using a Barnstead GenPure water purification system (ThermoScientific, Waltham, MA, USA). Bond Elute C18 cartridges (50 mg per 1 mL) were purchased from Agilent (Agilent Inc., Santa Clara, CA, USA). Mouse blood and plasma was purchased from Equitech-Bio, Inc. (Kerrville, TX, USA). All other materials and reagents were purchased from standard chemical suppliers and of analytical grade or higher.

### 4.2. LC-MS/MS, Stock and Sample Preparation 

#### 4.2.1. Liquid Chromatographic and Mass Spectrometric (LC-MS/MS) Conditions for MP1

An LC-MS/MS 8060 system (Shimadzu Scientific Instruments, Columbia, MD, USA) equipped with a dual ion source (DUIS) was used to perform mass spectrometric detection. LabSolutions LC-MS software Version 5.8 (Shimadzu Scientific, Inc., Columbia, MD, USA) was used for data acquisition and quantitation. To achieve the desired sensitivity, the compound dependent mass spectrometer parameters, such as temperature, voltage, gas pressure, etc., were optimized by auto method optimization via product ion search for MP1 and IS using a 0.5 µg/mL solution in MeOH. All analytes were detected in ESI− with the mass spectrometer source settings as follows: nebulizer gas: 2.0 L/min; drying gas: 10 L/min; heating gas: 10 L/min; interface temperature: 300 °C; desolvation line (DL) temperature: 250 °C; heat block temperature: 400 °C. The MS/MS system was operated at unit resolution in the multiple reaction monitoring (MRM) mode, using precursor ion > product ion combinations of 324.10 > 168.30 *m*/*z* for MP1 and 411.95 > 224.15 *m*/*z* for IS. The final optimum MS parameters are shown in Table 11.

All chromatographic separations were performed with an Acquity UPLC^®^BEH C18 column (1.7 µm, 100 × 2.1 mm, Waters, Inc. Milford, MA, USA) equipped with an Acquity UPLC C18 guard column (Waters, Inc. Milford, MA, USA). The mobile phase consisted of 0.1% acetic acid in water (mobile phase A) and methanol (MeOH) (mobile phase B) operated in isocratic mode with 10% A and 90% B with a total flow rate of 0.25 mL/min. The total run time was 6 min. The injection volume of all samples was 2 μL.

#### 4.2.2. Preparation of Stock, Calibration Standards and Quality Control Samples

Stock solutions of MP1 and IS were prepared at 1 mg/mL concentrations in DMSO. The stock solutions were diluted with MeOH to make working standard solutions and used to prepare the calibration standards (CSs) and quality control samples (QCs). CSs and QCs were prepared by spiking 10 µL of working standard solution and 10 µL of IS working solution (0.5 µg/mL) into 100 µL of blank mouse plasma or tissue homogenate to obtain a concentration range of 0.2–500 ng/mL. The final CS concentrations were: 0.2, 0.5, 1, 5, 10, 50, 200 and 500 ng/mL in bio-matrices. The QC comprised four concentrations: 0.2 ng/Ml—LLOQ, 0.6 ng/mL—LQC, 100 ng/mL—MQC, and 375 ng/mL—HQC. QCs were prepared separately in five replicates, independent of the calibration standards. CSs, QCs and all the main stock solutions were freshly prepared prior to use. Intermediate stocks, spiking calibration, and QC stock solutions were kept at −20 °C.

#### 4.2.3. Plasma, Tissue, Microsomes, and Liver S9 Fraction Sample Preparation

All analytes were extracted from CSs, QCs, mouse blood, and tissue samples by SPE using Agilent bond Elute C18 cartridges. Samples were prepared by spiking 10 µL of appropriate calibration stock in 100 μL blank bio-matrix (blank mouse plasma or tissue homogenate) and 10 µL of IS solution (0.5 µg/mL) and diluted with 600 µL of 1% acetic acid. A Cerex solid-phase extraction manifold from Varian (Palo Alto, CA, USA) with nitrogen to modulate flow was used for sample processing. Samples were vortexed for 2 min and then loaded onto SPE cartridges pre-conditioned with MeOH (2 mL), followed by water (1 mL). Loaded cartridges were washed with 15% MeOH (1 mL) and eluted with MeOH (2 mL). The eluate was evaporated under vacuum at room temperature then reconstituted in 200 µL of mobile phase (10:90; 0.1% AA: MeOH) then vortexed for 30 s and centrifuged at 13,400× *g* for 10 min at 4 °C, and the was supernatant transferred to HPLC vial for injection and analysis.

Each tissue sample was accurately weighed and then homogenized with de-ionized water at a 5- fold dilution factor for liver, spleen, brain, lungs, and kidney using a TissueLyserII (Qiagen Science, KY). Heart tissue was homogenized at a 6-fold dilution factor. The resultant tissue homogenate (100 µL) was spiked with 10 µL of IS solution and prepared for SPE as describe above for plasma samples.

S9 fraction and microsomal incubation buffer samples were extracted using simple protein precipitation. S9 fraction samples (100 µL) were obtained and mixed with 300 µL of ice-cold ACN spiked with 10 µL of IS (0.5 µg/mL) in a 1.5 mL-centrifuge tube. The samples were vortexed for 2 min, then centrifuged 17,940× *g* for 10 min. A 100 µL aliquot of the clear supernatant was transferred to an HPLC vial and 2 µL injected into the LC-MS/MS system.

### 4.3. Method Validation

The LC–MS/MS method was validated in accordance with the Food and Drug Administration (FDA) guidelines for Bioanalytical Method Validation [13] with respect to selectivity, specificity, linearity, accuracy, precision, matrix effect, stability, and dilution response. 

#### 4.3.1. Selectivity and Sensitivity 

Assay selectivity and specificity was determined by comparing the chromatogram of six different blank mouse plasma or tissue homogenate samples with that MP1 and IS-spiked plasma or tissue homogenate samples. Peak interference at the retention time of analytes and internal standard was investigated by comparing analyte response of interference-free blank plasma, plasma spiked with MP1 and the IS, and neat stock solutions in reconstitution solvent as described in Section 2.2.3.

The sensitivity of the method was determined from the signal-to-noise ratio (S/N) of the response of MP1 to blank matrix. The S/N ratio was calculated by comparing the baseline noise at the vicinity of the retention time of analyte with that of the LLOQ peak response. The lower limit of detection (LLOD) and LLOQ were defined as a three- and ten-fold S/N ratio, respectively. 

#### 4.3.2. Calibration Curve and Linearity

The calibration curves were established by a linear plot of the relation between the peak area ratio (MP1/IS) on the *y*-axis versus concentration of MP1 on the *x*-axis. The calibration curve was generated using weighted linear regression (1/x^2^). Each calibration curve consisted of a blank sample, a zero sample (blank + IS), and 12 non-zero concentrations consisting of eight CSs and four QCs. The acceptance criteria for each CS concentration was ±15% standard deviation (SD) from the nominal value, except at LLOQ, which was set at ±20%. The calibration curve had to have a correlation coefficient (r^2^) of 0.988 or better for MP1. The assay was linear over the range of 0.2–500 ng/mL.

#### 4.3.3. Carry-Over

Residual carry-over was assessed by injecting two consecutive “zero samples” after running the HQC (375 ng/mL). The response of the first zero sample was required to be <20% of the response of a processed LLOQ sample to conclude no carry-over.

#### 4.3.4. Accuracy and Precision 

Intra and inter–assay accuracy and precision were evaluated by analyzing five replicates of QC samples at four different concentrations (LLOQ, LQC, MQC, and HQC) in mouse plasma or tissue homogenate for three consecutive days. Precision was defined as the percent relative standard deviation (%RSD) with acceptance criteria of ±15% (except ±20% at LLOQ). Accuracy was defined as the percent bias (% Bias) with the same acceptance criteria as precision. %bias was calculated according to the following Equation (1): (1)$$\%\mathrm{Bias}=\frac{\left(\mathrm{observed}\;\mathrm{conc}.-\mathrm{nominal}\;\mathrm{conc}.\right)}{\mathrm{nominal}\;\mathrm{conc}}\times 100$$

#### 4.3.5. Extraction Recovery and Matrix Effect

Extraction recovery was calculated at three different QC concentrations (LQC 0.6 ng/mL), (MQC 100 ng/mL) and (HQC 375 ng/mL) for both MP1 and IS (0.5 µg/mL) by comparing the mean peak area of an analyte spiked before extraction to the mean peak area of an analyte spiked post-extraction multiplied by 100. 

To assess the effect of the matrix on analyte response, blank mouse plasma and tissue homogenate were processed as described and the dry post-extract was spiked with analyte prepared equivalent to the QCs. The average peak area from analytes spiked in the extracted blank matrix was compared to the response for QCs prepared in reconstitution solvent. The absolute matrix effect (ME) and IS normalized ME were calculated as described in Equations (2) and (3).
(2)$$\mathrm{ME}=\frac{\mathrm{Mean}\;\mathrm{Peak}\;\mathrm{area}\;\mathrm{of}\;\mathrm{analyte}\;\mathrm{spiked}\;\mathrm{post}-\mathrm{extraction}}{\mathrm{Mean}\;\mathrm{Peak}\;\mathrm{area}\;\mathrm{of}\;\mathrm{analyte}\;\mathrm{in}\;\mathrm{Solvent}}\times 100$$
(3)$$\mathrm{IS}\;\mathrm{normalized}\;\mathrm{ME}=\frac{\mathrm{Mean}\;\mathrm{Peak}\;\mathrm{area}\;\mathrm{ratio}\;\mathrm{of}\;\mathrm{analyte}/\mathrm{IS}\;\mathrm{spiked}\;\mathrm{post}-\mathrm{extraction}}{\mathrm{Mean}\;\mathrm{Peak}\;\mathrm{area}\;\mathrm{ratio}\;\mathrm{of}\;\mathrm{analyte}/\mathrm{IS}\;\mathrm{in}\;\mathrm{Solvent}}\times 100$$

#### 4.3.6. Stability

Storage stability of MP1 in mouse plasma samples was determined at three different QC concentrations. The following stabilities were evaluated: bench-top storage (4 h at room temperature 21 °C), three freeze–thaw cycles (−80 °C to room temperature for 30 min, back to −80 °C, stored for 24 h, and repeated twice more), and long-term storage (12 months at −80 °C). Additionally, 24 h auto-sampler stability of extracted samples (at 4 °C) was determined. The sample concentration for MP1 under each condition was tested and mean values for accuracy and precision calculated. 

#### 4.3.7. Dilution Integrity

The dilution integrity of samples was tested on six replicates of three levels of dilution: two- (2× HQC), five- (5× HQC), and ten-fold (10× HQC) dilutions of the high QC concentration. The calculated concentration measurements were compared to the nominal concentration at each dilution level. 

#### 4.3.8. Solubility of MP1

The solubility of MP1 was assessed by spiking 1% DMSO stock solutions (5 µL) into 100 mM phosphate buffered saline (pH 7.4, 445 µL) in triplicate to produce final assay concentrations of 1, 5, 10, 20, 40, 50, 60, 80 µM. Spiked samples were kept at room temperature for 2 h. At pre-determined times, each microcentrifuge tube was centrifuged at 10,000× *g* for 15 min. Supernatant was aspirated and mixed with equal volume of acetonitrile. Calibration curve samples were prepared in buffer, acetonitrile, DMSO media for analysis. Solubility concentration was calculated via the calibration curve using analyte/IS peak area ratio and concentration at which the analyte precipitated out, which was considered to be the upper soluble concentration. 

### 4.4. In-Vitro Studies

#### 4.4.1. Gastrointestinal Fluid Stability Studies of MP1

Simulated gastrointestinal fluids were prepared according to USP specifications. Briefly, SGF was formulated consisting of sodium chloride (0.2 g), pepsin (3.2 g), hydrochloric acid (7 mL) in water (1 L, pH 1.2). SIF was formulated with potassium dihydrogen phosphate (6.805 g), sodium hydroxide (0.896 g), pancreatin (10 g) in water (1 L, pH 6.8). MP1 at 10 µM concentration was incubated with SGF, SIF and phosphate buffer (100 mM, pH 7.4) at 37 °C on a shaking water bath each in triplicate. Samples (200 µL) were collected at 0, 15, 30, 60 and 120 min. Immediately after sample collection, five volumes of MeOH containing IS were added and samples prepared for analysis. Stability was determined as a percentage parent compound detected at each time point relative to the concentration at 0 min in buffer (100%) [23].

#### 4.4.2. The Blood to Plasma Ratio (Kb/p) 

The mouse blood to plasma partitioning of MP1 was determined at concentrations of 1 µM. Fresh mouse blood (800 µl) was incubated in a water bath maintained at 37 °C for 10 min. After incubation, blood was spiked with the drug from stock solutions of 1 µM (325 ng/mL) to achieve final concentrations and to maintain organic content <1%. At different time points (0, 30, 60 min), aliquots (50 µL) were collected for blood analysis and an additional sample (120 µL) was transferred into a micro-centrifuge tube and centrifuged at 4000× *g* for 10 min at 4 °C to separate 50 µL plasma for analysis. Blank plasma (50 µl) was added to the blood aliquot, and 50 µL of blank blood was added to the separated plasma for matrix match and calibration in 100 µL. The whole blood and plasma samples were further processed by SPE as described above. The analyte peak areas ratios were used to calculate the blood-to-plasma (B/P) ratio using Equation (4).
(4)BPRatio=Analyte Concentration in whole bloodAnalyte Concentration in Plasma

#### 4.4.3. Plasma Protein Binding (PPB) Study

A Rapid Equilibrium Dialysis (RED) device system (Thermo Scientific, Rockford IL) was used to evaluate PPB by adding buffer to the buffer chamber and dosing solution to the sample chamber. The buffer chamber contained 350 µL of phosphate-buffered saline (containing 100 mM sodium phosphate and 140 mM sodium chloride, pH 7.4). The sample contained mouse plasma spiked with 1 µM or 10 µM MP1 and was added to the sample chamber. The RED kit top was then sealed and incubated at 37 °C on an orbital shaker at 100 rpm for 5 h.

Plasma stability of MP1 was assessed along with the non-specific binding potential of MP1 to the dialysis membrane and cell chambers by spiking drug into buffer while incubating at 37 °C for 5 h and comparing it with 0 h concentration. At pre-determined times, aliquots (40 µL) were removed from the sample and buffer chambers and mixed with an equal volume of buffer or blank plasma, respectively. The samples were further processed by SPE as described above and analyzed by LC-MS/MS. The stability, equilibrium, device recovery and PPB were calculated using following equations [21].
(5)$$\%\;Remaining\;at\;5\;h=\frac{Concentration_{4\;h}}{Concentration_{0\;h}}\times 100$$
(6)$$\%\;Equilibrium=\frac{Concentration_{receiver\;cell}}{Concentration_{donor\;cell}}\times 100$$
(7)$$\%\;Device\;Recovery=\frac{Concentration_{receiver\;cell}+Concentration_{donor\;cell}}{Concentration_{4\;h}}\times 100$$
(8)$$\%\;Bound=\frac{Concentration_{donor\;cell}-Conentration_{receiver\;cell}}{Concentration_{donor\;cell}}\times 100$$

#### 4.4.4. In-Vitro Metabolic Stability in Mouse S9 Fraction

In-vitro phase I and II metabolism were assessed using mouse liver S9 fractions (XenoTech, LLC, Lenexa, KS, USA). Briefly, 1 µM MP1 was pre-mixed with a solution mixture of potassium phosphate buffer (100 mM, pH 7.4) at 37 °C containing the following: 1 mM NADPH, 4 mM saccharolactone, 1 mM uridine 4′-diphospho-glucuronic acid, and 0.1 mM 3′-phosphoadenosin-4′-phosphosulphate. Incubation was performed in triplicate, *n* = 3. Reactions were performed at a final incubation volume of 1000 µL and initiated with the addition of the S9 fraction, 50 µL of 20 mg/mL protein concentration. Immediately after mixing the MP1 into the incubation mixture, a sampling point was taken, representing t = 0. Subsequent sampling points were performed at the following intervals: 5, 15, 20, 30, 45, 60, 90 and 120 min. To stop the reaction, 100 µL of the sample were added to 1.5 mL-centrifuge tube containing 300 µL of ice-cold ACN spiked with 10 µL of IS (0.5 µg/mL). All the samples were vortexed and centrifuged at 13,000× *g* for 10 min. The resultant supernatant was transferred to an autosampler vial and 2 µL injected on the LC-MS/MS. Testosterone, 7-HC, and diclofenac were utilized as positive controls to ensuring proper incubation conditions were maintained.

#### 4.4.5. In-Vitro Metabolic Stability in Mouse, Rat, and Human Liver Microsomes

Metabolic stability was assessed using mouse, rat, and human liver microsomes (XenoTech, LLC, Lenexa, KS, USA) for phase I metabolism. Briefly, the buffer solution was prepared containing potassium phosphate buffer (100 mM, pH 7.4), 25 µL of microsomal protein (20 mg/mL), magnesium chloride (10 mM) and NADPH (2 mM) in a final volume of 0.5 mL was pre-incubated at 37 °C for 10 min in water bath maintaining at 60 rpm. The reaction was started by adding 2µL of MP1 (1 µM). Serial samples (50 µL) were collected at selected time intervals (0, 5, 15, 20, 30, 45 and 60 min) and quenched with 200 µL of acetonitrile spiked with 10 µL of IS (0.5 µg/mL). All the samples were vortexed and centrifuged at 13,000× *g* for 15 min, supernatant collected and transferred to an autosampler vial and injected (2 µL) onto the LC-MS/MS system. Testosterone and diclofenac were used as positive controls to ensure that HLM and incubation conditions were appropriate to conduct metabolism studies.

Both the S9 Fraction and HLM metabolic stability was expressed as the percentage of drug remaining at each time point. The in-vitro metabolic elimination rate constant was calculated from the first-order plot of a natural logarithm of the area ratio versus time. The slope of the linear regression equation was used to determine elimination rate constant “k” (min^−1^). The half-life (t_1/2_) was calculated using Equation (9). The in-vitro intrinsic clearance (CL_int_) was determined by using Equation (10). The intrinsic clearance was further extrapolate to in-vitro hepatic clearance (CL_int,H:_ mL/min/kg of body weight) using a scaling factors and Equation (11) [24,25].
(9)$$t_{1/2}=0.693/k$$
(10)$${\mathrm{CL}}_{\mathrm{int}}=\frac{0.693}{t_{1/2}}\times \frac{\mathrm{Volume}\;\mathrm{of}\;\mathrm{reaction}\;\mathrm{mixture}\;\left(\mathrm{mL}\right)}{\mathrm{mg}\;\mathrm{of}\;\mathrm{protein}}$$
(11)$${\mathrm{CL}}_{\mathrm{int},H}={\mathrm{CL}}_{\mathrm{int}}\times \frac{\mathrm{mg}\;\mathrm{of}\;\mathrm{protein}}{\mathrm{gram}\;\mathrm{of}\;\mathrm{liver}}\times \frac{\mathrm{gram}\;\mathrm{of}\;\mathrm{liver}}{\mathrm{kg}\;\mathrm{of}\;\mathrm{body}\;\mathrm{weight}}$$

#### 4.4.6. Metabolite Identification

To identify MP1 metabolites, we performed a mass shift analyses using the same LC-MS/MS system. A 35 min gradient time profile was used for metabolite identification via Q1 scan mode, which ranged of 250 to 1000 Da. The gradient profile of the mobile phase was held at 10% mobile phase B (MeOH) for 1 min then increased up to 95% mobile phase B over 28 min, then brought back to 10% B over 1 min, followed by a 4 min equilibration. The parent drug MS/MS spectrum was used as a template model for metabolites structural identification and assuming the fragmentation pattern of the parent drug, MP1, from the product-ion spectrum, which was used to deduce the structures of the metabolites [11,22]. The data acquisition for metabolite identification study was conducted using LabSolutions, version 5.8 (Shimadzu Scientific Inc, Columbia, MD, USA)

### 4.5. In-Vivo Pharmacokinetic Studies

#### 4.5.1. Biodistribution Studies Animals, Drug Administration and Sampling

The University of Nebraska Medical Center Institutional Animal Ethics Committee approved all animal studies (protocol number 17-046-06-FC). Female BALB/c mice, with weights ranging from 24 to 30 g were used for biodistribution studies. Animals were housed in the University of Nebraska Medical Center animal facility, for at least 7 days prior to the experiments, in order to acclimatize the animals to the laboratory conditions, at a temperature of 23–24 °C, relative humidity of 40–70% and 12/12 h light/dark cycles. The dosing solution was made of DMSO-Polyethylene glycol 400 (PEG400)-Propylene glycol (PG)-EtOH-Cremophore-PBS (2/20/10/10/5/53% *v*/*v*). MP1 (15 mg/kg) was administered by oral gavage, the dose was selected based on previous pharmacological reports in mice [7]. After dosing, approximately 100 µL of blood was collected in polypropylene tube, from the maxillary vein at 5, 15, 30 min and 1, 2, 4, 6, 8, 24, 48 h. Two blood time points was collected from every mouse and third time point was the terminal time point, for a total of three-time points from each mouse (5 mice/group/per time point). Plasma was prepared by centrifugation at 4000× *g* at 4 °C for 10 min. The collected plasma samples were stored at −80 °C until analysis.

Tissues (liver, lungs, heart, kidney, brain, and spleen) were collected at 2, 4, 8- and 48-h following dosing. Tissue samples were rinsed with phosphate buffered saline to remove blood and then blotted with filter paper. After weighing, each tissue sample was individually homogenized with de-ionized water, 5-dilution factor for liver, spleen, brain, lungs, and kidney while the heart was homogenized at a 6-dilution factor using a TissueLyser II (Qiagen Science, KY) then all tissues were stored at −80 °C until analysis. Plasma concentrations (ng/mL) and tissue concentrations (ng/g) were determined for each time point collected. Drug accumulation in tissue was determined by calculating a tissue to plasma concentration ratio (Kp) for each tissue.

#### 4.5.2. Data Analysis

Non-compartmental pharmacokinetics analysis was performed on Phoenix^®^ 8.2 (Certara Corporation, Mountain View, CA, USA) to determine plasma pharmacokinetic parameter of MP1 in plasma. Peak plasma concentration (C_max_) and time for the peak plasma concentration (T_max_) was obtained from visual inspection of the concentration-time plot. The area under the curve (AUC_0–∞_) was estimated using the linear trapezoidal method from 0–t_last_ and extrapolation from last time point to infinity based on the observed concentration at the last time point divided by the terminal elimination rate constant (k).

The elimination half-life (t_1/2_) was calculated using Equation (12). Clearance (CL) was calculated using Equation (13) and the apparent volume of distribution of the elimination phase (V_d_) were calculated using Equation (14). The tissue to blood (Kp) ratio was calculated by using Equation (15).
(12)$$t_{1/2}=\frac{0.693}{k}$$
(13)$$\mathrm{CL}=\frac{\mathrm{Dose}\;\left(\frac{\mathrm{mg}}{\mathrm{kg}}\right)}{{\mathrm{AUC}}_{0–\infty }}$$
(14)$$V_{d}=\frac{\mathrm{Dose}\;\left(\frac{\mathrm{mg}}{\mathrm{kg}}\right)}{{K*\mathrm{AUC}}_{0–\infty }}$$
(15)$$K_{P}=\frac{\mathrm{Concentration}\;\mathrm{in}\;\mathrm{Tissue}}{\mathrm{Concentration}\;\mathrm{in}\;\mathrm{Blood}}$$

## 5. Conclusions

In conclusion, we successfully developed and validated a rapid, selective, and sensitive LC-ESI-MS/MS method for the quantitation of MP1 in mouse plasma and tissues. The assay was linear over the concentration range of 0.2–500 ng/mL. The LC–MS/MS bioanalytical method was validated according to FDA guidelines, showed acceptable selectivity, no significant matrix effects, and acceptable accuracy and precision. In addition, we applied the validated method to the assessment of MP1 in-vitro and in-vivo ADME properties in biological samples with nominal concentrations as low as 0.2 ng/mL.

## Figures and Tables

**Figure 1 molecules-25-05898-f001:**
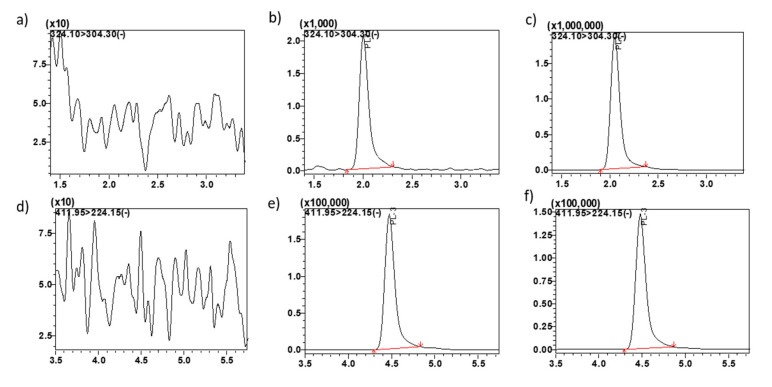
Representative MRM ion-chromatograms. (**a**) blank mouse plasma using the conditions for MP1 detection, (**b**) MP1 spiked in mouse plasma at LQC 0.6 ng/mL, (**c**) mouse plasma from pre-clinical study sample at 0.5 h time point showing MP1, (**d**) blank plasma using the conditions for PL3 (IS) detection, (**e**) PL3 (IS) spiked in plasma 0.5 µg/mL, (**f**) plasma from pre-clinical study sample at 0.5 h time point spiked with IS showing PL3.

**Figure 2 molecules-25-05898-f002:**
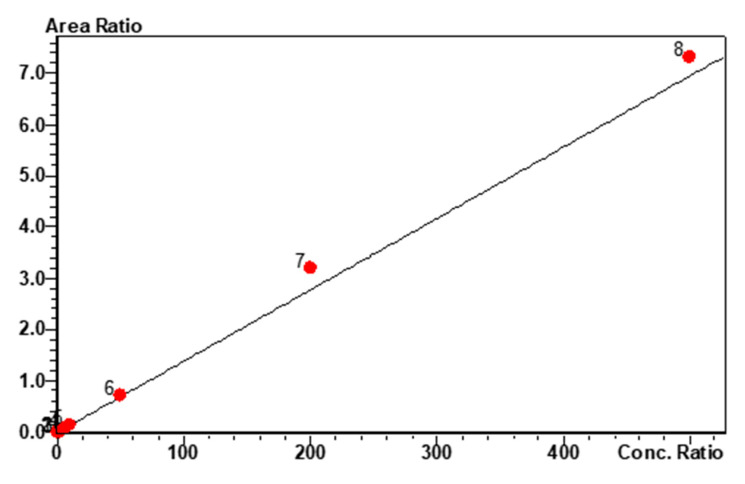
MP1 calibration curve and linearity. Linear calibration fit for MP1 over the concentration range of 0.2 to 500 ng/mL, r^2^ = 0.988 and %RSD = 9.97.

**Figure 3 molecules-25-05898-f003:**
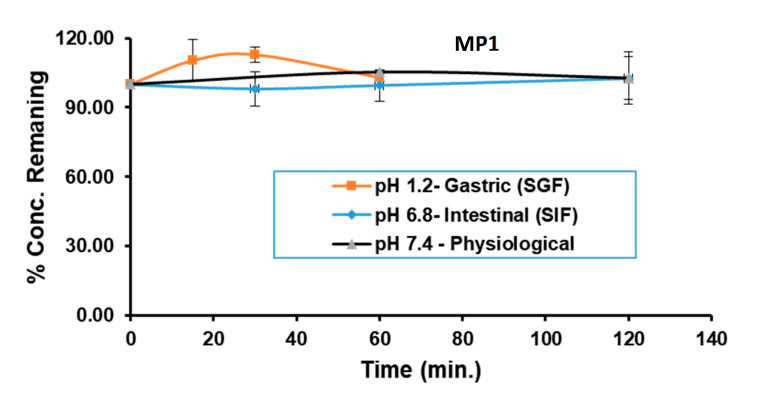
Gastrointestinal Stability of MP1. MP1 stability following a 1 h incubation in SGF (pH 1.2), a 2 h incubation in SIF (pH 6.8) and in buffer (pH 7.4).

**Figure 4 molecules-25-05898-f004:**
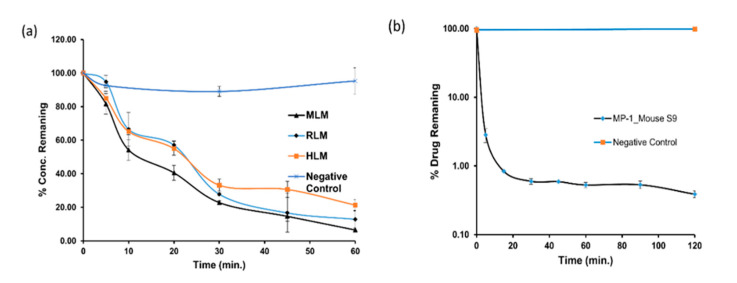
MP1 In-vitro metabolic stability in microsomes. Time-dependent metabolic depletion (% turnover or amount of drug remaining vs. incubation time) of MP1 in microsomes and mouse S9 fraction (**a**) interspecies microsomal metabolic stability of MP1 in presence of NADPH and absence of NADPH as a negative control (**b**) mouse S9 metabolic stability of MP1 in presence of NADPH and absence of NADPH as a negative control. Data shown as mean ± SD (*n* = 3).

**Figure 5 molecules-25-05898-f005:**
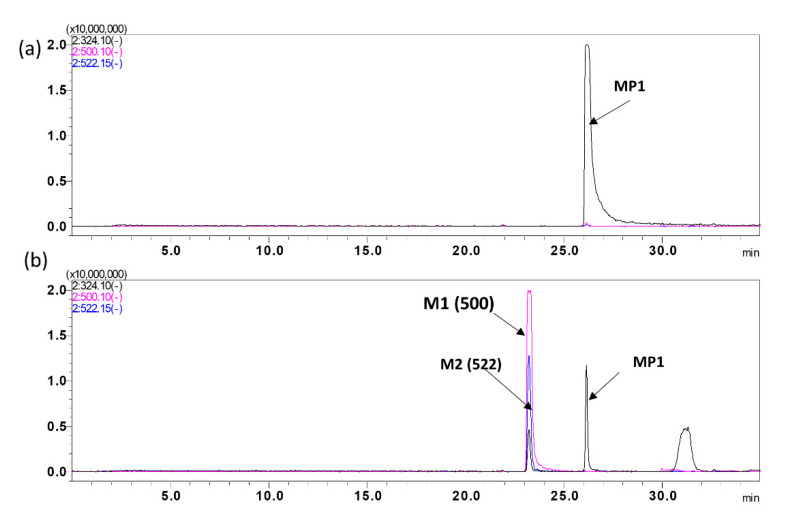
MP1 Metabolite identification. Representative overly chromatogram of MP1 and its metabolites, utilizing a 35 min gradient for separation after, (**a**) 0 min incubation, and (**b**) 90 min incubation.

**Figure 6 molecules-25-05898-f006:**
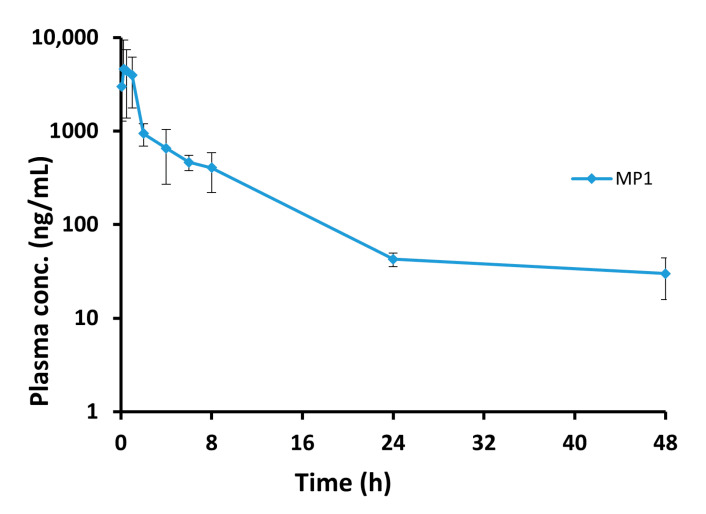
The blood concentration vs. time profile for the MP1. Plasma concentration-time profile after 15 mg/kg oral administration of MP1 (Mean ± SD, *n* = 4 at each time).

**Figure 7 molecules-25-05898-f007:**
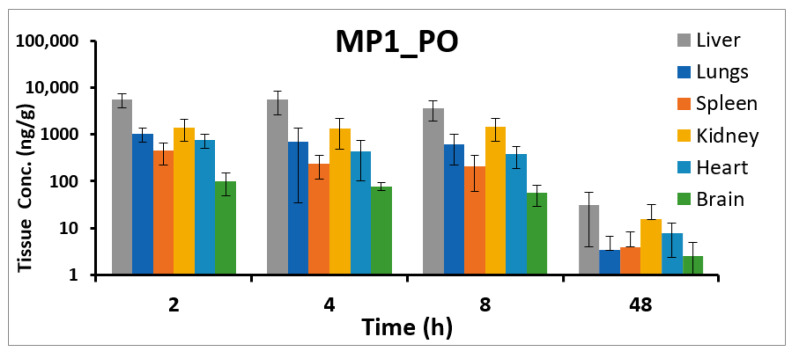
MP1 tissue concentrations. Tissue concentrations (ng/g) at different time points in liver, lungs, heart, kidney, brain, and spleen after 15 mg/kg oral administration of MP1 in mice (Mean ± SD, *n* = 4).

**Table 1 molecules-25-05898-t001:** MP1 accuracy and precision. Intra–assay and inter–assay accuracy and precision of MP1 in mouse plasma (*n* = 6).

Nominal Conc. (ng/mL)	Accuracy		Precision	
	%Bias Intra–Assay	%Bias Inter–Assay	%RSD Intra–Assay	%RSD Inter–Assay
LLOQ (0.2 ng/mL)	7.3	4.8	10.8	8.3
LQC (0.6 ng/mL)	3.9	−2.7	4.4	12.9
MQC (100 ng/mL)	−4.7	−14.0	1.4	13.6
HQC (375 ng/mL)	−13.4	−12.0	5.7	4.4

**Table 2 molecules-25-05898-t002:** Recovery and matrix effect. Assessment of the recovery and matrix effect of MP1 in mouse plasma, (Mean ± SD, *n* = 3).

Nominal Conc. (ng/mL)	% Extraction Recoveries (Mean ± SD, *n* = 3)	% Matrix Effect (Mean ± SD, *n* = 3)
LQC (0.6 ng/mL)	95.8 ± 2.6	103.7 ± 8.6
MQC (100 ng/mL)	89.1 ± 10.6	90.5 ± 4.9
HQC (375 ng/mL)	90.8 ± 3.4	99.6 ± 7.6
Internal standard (IS) (0.5 ng/mL)	87.3 ± 3.8	105.1 ± 11

**Table 3 molecules-25-05898-t003:** Stability of MP1. Stability was tested in mouse plasma at different storage conditions, (Mean ± SD, *n* = 3).

Nominal Conc. (ng/mL)	Measured Mean Conc. (ng/mL)	% Accuracy
Autosampler stability 4 °C, Mean ± SD, *n* = 3.		
LQC (0.6 ng/mL)	0.515 ± 0.107	85.8 ± 17.8
MQC (100 ng/mL)	102.8 ± 9.5	102.85 ± 9.45
HQC (375 ng/mL)	340.6 ± 12.8	90.8 ± 3.4
Bench-top stability 21 °C, Mean ± SD, *n* = 3.		
LQC (0.6 ng/mL)	0.57 ± 0.03	94.2 ± 4.9
MQC (100 ng/mL)	114.8 ± 2.5	114.9 ± 2.6
HQC (375 ng/mL)	386.5 ± 15.2	103 ± 4
Freeze-thaw stability −80 °C, up to 3 Cycle, Mean ± SD, *n* = 3.		
LQC (0.6 ng/mL)	0.67 ± 0.06	108.6 ± 8.9
MQC (100 ng/mL)	102.8 ± 11.95	101.4 ± 10
HQC (375 ng/mL)	366.3 ± 44.9	97.7 ± 11.96
Long term stability −80 °C, 12 months, Mean ± SD, *n* = 3.		
LQC (0.6 ng/mL)	0.61± 0.02	100.3 ± 4.1
MQC (100 ng/mL)	92.40 ± 7.92	92.4 ± 7.9
HQC (375 ng/mL)	383.69± 79.94	101.8 ±13.5

**Table 4 molecules-25-05898-t004:** The value MP1 the blood to plasma ratio (Kb/p). MP1 concentration in blood and plasma, (Mean ± SD, *n* = 3).

Time (min)	Blood Conc.	Plasma Conc.	
	Mean (ng/mL) ± SD	Mean (ng/mL) ± SD	Kb/p
0	294.1 ± 11.1	727.73 ± 49.16	0.40
30	490.7 ± 83.0	725.84 ± 157.67	0.68
60	490.8 ± 29.8	781.54 ± 24.53	0.63

**Table 5 molecules-25-05898-t005:** MP1 Plasma Protein Binding. Plasma protein binding in mouse plasma (Mean ± SD values for MP1, *n* = 3).

Nominal Conc. (µg/mL)	% Plasma Protein Bound ± SD
MP1 (1 µg/mL)	99.96 ± 0.03
MP1 (10 µg/mL)	99.97 ± 0.02

**Table 6 molecules-25-05898-t006:** In-vitro metabolic stability in mouse S9 fraction. MP1 estimates for CL_int_, t_½_ and CL_int,H_ in mouse (MLM), rat (RLM) and human (HLM) liver microsomes, and mouse S9, (Mean ± SD, *n* = 3).

Parameters	MLM	RLM	HLM	Mouse S9
t_½_ (min)	15.64 ± 0.46	18.1± 3.73	27.47 ± 2.68	4.9 ± 0.2
CL_int_ (µL/min/mg protein)	88.72 ± 2.66	80.01 ± 16.5	50.99 ± 5.25	142.3 ± 5.6
CL_int,H_ (mL/min/kg wgt)	479.07 ± 14.36	288.02 ± 59.41	81.58 ± 8.39	N/A

**Table 7 molecules-25-05898-t007:** MP1 metabolites chromatogram summary. MP1 metabolites biotransformation, mass shift, precursor ion and retention time.

Peak ID	Biotransformation	Mass Shift	Precursor Ion (*m*/*z*)	Retention Time (Minutes)
	Parent (MP1)	0	324.10	26.7
M1	Glucuronidation	176	500.12	23.0
M2	Unknown	198	522.15	23.0

**Table 8 molecules-25-05898-t008:** MP1 plasma concentrations following oral administration. (15 mg/kg) to mice (Mean ± SD, *n* = 4).

Time (h)	Mean Plasma Conc. (ng/mL)	SD
0.08	2990.2	1705.6
0.25	2452.1	313.7
0.50	4412.7	3038.8
1	3985.7	2222.2
2	943.2	251.6
4	655.3	385.4
6	465.8	87.1
8	405.9	184.5
24	42.7	7.1
48	30.1	14.2

**Table 9 molecules-25-05898-t009:** Summary of MP1 pharmacokinetic parameters. Pharmacokinetic parameters of MP1 after 15 mg/kg oral administration (Mean, SD, SE and CV%, *n* = 20).

PK Parameter	Mean	SD	SE	CV%
C_max_ (ng/mL)	4714.7	±2343.5	1171.8	49.7
T_max_ (h)	0.6	±0.4	0.2	68.9
t_1/2_ (h)	9.2	±1.7	0.9	18.7
AUC_0–∞_ (h × ng/mL)	15367.8	5466.9	2733.4	35.6
AUC_0–last_ (h × ng/mL)	14951.7	±5357.2	2678.6	35.8
V_d_ (L/kg)	13.9	±4.3	2.1	30.9
CL (L/h/kg)	1.1	±0.3	0.2	31.6

**Table 10 molecules-25-05898-t010:** MP1 tissue concentrations. Tissue concentrations (ng/g) in liver, lungs, heart, kidney, brain, and spleen after 15 mg/kg oral administration of MP1 in mice (Mean ± SD, *n* = 4 at each time).

Time (h)	Lung		Spleen		Kidney		Brain		Heart		Liver	
	Mean (ng/g)	SD	Mean (ng/g)	SD	Mean (ng/g)	SD	Mean (ng/g)	SD	Mean (ng/g)	SD	Mean (ng/g)	SD
2	1036.2	338.0	445.4	219.9	1432.2	715.8	100.1	50.8	759.1	258.1	5613.8	1941.2
4	705.3	671.4	236.3	126.8	1351.6	874.8	77.7	14.9	433.0	331.3	5583.3	2912.9
8	619.2	392.5	206.6	145.2	1463.7	751.2	56.3	27.5	376.2	187.9	3615.3	1699.5
48	3.4	3.4	3.9	4.3	15.3	16.3	2.5	2.4	7.6	5.3	30.6	26.6

**Table 11 molecules-25-05898-t011:** Summary of MS/MS parameters: precursor ion, fragment ions, voltage potential (Q1), collision energy (CE) and voltage potential (Q3) for MP1 and IS.

Analytes	MRM Transition *m*/*z* (Q1 → Q3)	Q1 (V)	Q3 (V)	CE (V)	Retention Time (min)
Target: MP1	324.10 > 168.30	−11	−10	−22	1.9
	324.10 > 304.30	−11	−20	−18	
Internal standard (IS): PL3	411.94 > 224.14	−14	−14	−28	4.5

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
