# Peer review of "Assessment of Tissue Distribution and Metabolism of MP1, a Novel Pyrrolomycin, in Mice Using a Validated LC-MS/MS Method"

_molecules, 2020, doi:10.3390/molecules25245898_

Round 1

Reviewer 1 Report

The authors present a LC-MS/MS method applied to the study of pharmacokinetics, specifically tissue distribution and metabolism of the MP-1 anticancer agent in mice. This is an interesting work and adds to the body of literature re MP-1. However, the authors should address the following comments major and minor comments. 

Major comments:

  •  title: pharmacokinetics includes distribution and metabolism assessment: this is redundant in the tile. 
  • Introduction: it provides a broad introduction on neuroblastoma, while falling short on the marine compounds object of the study (MP-1). We suggest to shorten the neuroblastoma related aspects and expand on the MP-1 compound, related literature and tie back to the associated lack of knowledge. 
  • Metabolite identification: did you observe any phase I biotransformation e.g. oxidation, hydroxylation? Extend the discussion and data on this point as possible. Were these metabolites validated in the plasma samples? 
  • Tissue distribution: did you observe accumulation in the fat tissues and if not, please discuss as the fat compartment is an important accumulation site. 

Minor comments:

  • Line 162: both 1.2 and 7.4 pH values are physiological, they differ just from the localization (stomac, gut, blood..) Please correct. 
  • Line 284-286: this is ok, may be worth mentioning that studies can also point to future opportunities to measure cellular penetration&concentration. 

Author Response

Reviewer 1

Comments and Suggestions for Authors

The authors present a LC-MS/MS method applied to the study of pharmacokinetics, specifically tissue distribution and metabolism of the MP-1 anticancer agent in mice. This is an interesting work and adds to the body of literature re MP-1. However, the authors should address the following comments major and minor comments. 

Major comments:

  • title: pharmacokinetics includes distribution and metabolism assessment: this is redundant in the tile. 
  • Author response: The correction has been made as per reviewer’s suggestion, “Pharmacokinetics” was deleted and the title changed to Assessment of tissue distribution and metabolism of MP1, a novel pyrrolomycin, in mice using a validated LC-MS/MS method.

  • Introduction: it provides a broad introduction on neuroblastoma, while falling short on the marine compounds object of the study (MP-1). We suggest to shorten the neuroblastoma related aspects and expand on the MP-1 compound, related literature and tie back to the associated lack of knowledge.
  • Author response: The correction has been made as per reviewer’s suggestion. The two neuroblastoma paragraphs ( Line 38- Line 49) was shortened to ( Line 38- 40 line ). We elaborated on the marinopyrroles (Lines 108-110).

  • Metabolite identification: did you observe any phase I biotransformation e.g. oxidation, hydroxylation? Extend the discussion and data on this point as possible. Were these metabolites validated in the plasma samples? 

Author response: We did not observe any phase I metabolites (oxidation, hydroxylation) in these initial studies. In S9 fraction, the disappearance rate of MP-1 is much greater compared to MLM, RLM and HLM S9 in the presence of UDPGA or NADPH, respectively (Figure 4).  In S9 fraction, because glucuronidation metabolite was found to be the dominant metabolite in a metabolite-profiling study, suggested the direct glucuronidation was the major clearance pathway. For confirmation detailed metabolism studies and   identification of CYP or UGT involvement in metabolism of MP-1, will be determined in future studies. We have clarified this in the discussion (Line 284- Line 290).

  • Tissue distribution: did you observe accumulation in the fat tissues and if not, please discuss as the fat compartment is an important accumulation site. 
  • Author response: We have not measured the drug accumulation in fat tissue. In this study we mainly focused on the accumulation of the drug in vital organs, specifically the Brain which we believe to be our site of action. We do aim to conduct additional studies like Physiologically based pharmacokinetic (PBPK) dosing which will have body composition as a cofactor and agree that determination of drug in the fat tissue will be critical for these studies.

Minor comments:

  • Line 162: both 1.2 and 7.4 pH values are physiological, they differ just from the localization (stomach, gut, blood..) Please correct. 
  • Author response: This section has been condensed and “physiological” removed (Line 166 to 169).

  • Line 284-286: this is ok, may be worth mentioning that studies can also point to future opportunities to measure cellular penetration & concentration.
  • Author response: Thank you for the comment, we agree with the reviewer and have added this to the discussion (line 297-300.)

Reviewer 2 Report

The authors developed a rapid, selective, and sensitive liquid chromatography coupled with tandem mass spectrometry (LC-MS/MS) method for determination of MP1 in mouse plasma. It provided a basis for analysis for MP1. There are problems should be modified.

  1. The abbreviation should be provided the full name in the first time.
  2. The abstract should be modified according to the magazine format
  3. In the introduction, lacks organization of content, brevity. So, please modify the introduction, let it is correct, clear and updated.
  4. “2.2. Plasma and tissue sample preparation and liver S9 fraction sample preparation” should be listed to 2.1.
  5. “2.3.1. Selectivity and Sensitivity” should be changed to “2.3.1. Selectivity and sensitivity”.
  6. Line 286, “Table 1. Summary of MS/MS parameters: precursor ion, fragment ions, voltage potential (Q1), collision energy (CE) and voltage potential (Q3) for MP1 and IS.”. Whats mean is it? It should be Table 11!
  7. The partial contents of in part of discussion should be listed in introduction.

Author Response

Reviewer 2:

The authors developed a rapid, selective, and sensitive liquid chromatography coupled with tandem mass spectrometry (LC-MS/MS) method for determination of MP1 in mouse plasma. It provided a basis for analysis for MP1. There are problems should be modified.

  • The abbreviation should be provided the full name in the first time.
  • Author response: The correction has been made as per reviewer’s suggestion. Line 38 Neuroblastoma changed to NB, Line 207 and Line 318 changed negative Ionization mode to ESI-, Line 80 and Line 253 electrospray ionization (ESI).

  • The abstract should be modified according to the magazine format
  • Author response: The correction has been made as per reviewer’s suggestion. The headings were deleted ( 1) background , 2) Methods, 3) Results, 4) Conclusion) .

  • In the introduction, lacks organization of content, brevity. So, please modify the introduction, let it is correct, clear and updated.
  • Author response: We have modified the introduction as per the reviewer 1 suggestions. The two neuroblastoma paragraph ( Line 38- Line 49) was shortened to ( Line 38- 40 line ). We elaborated on the marrinopyrrole compounds ( Line 59- 60).

  • “2.2. Plasma and tissue sample preparation and liver S9 fraction sample preparation” should be listed to 2.1.
  • Author response: Thank you for your careful observations. We have made the appropriate changes to our manuscript and section 2.2 Plasma and tissue sample preparation and liver S9 fraction sample preparation merge to section 2.2.

  • “2.3.1. Selectivity and Sensitivity” should be changed to “2.3.1. Selectivity and sensitivity”.
  • Author response: The correction has been made as per reviewer’s suggestion. In addition, we have modified additional sections: Line 104, Line 165 (Gastrointestinal stability), Line 178 ( 2.4.3. Plasma Protein Binding (PPB) study), Line 190 (2.4.5. In vitro metabolic stability in mouse S9 fraction) and Line 465 (4.4.4. In vitro metabolic stability in mouse S9 fraction),

  • Line 286, “Table 1. Summary of MS/MS parameters: precursor ion, fragment ions, voltage potential (Q1), collision energy (CE) and voltage potential (Q3) for MP1 and IS.”. Whats mean is it? It should be Table 11!
  • Author response: Thank you for bringing these errors to our attention. We have made the appropriate changes and Yes, it should be Table 11 not Table 1.
  • The partial contents of in part of discussion should be listed in introduction.
  • Author response: We are not clear what is being asked for this comment but have modified the introduction and discussion for clarity.

Round 2

Reviewer 1 Report

The authors addressed my concerns. The manuscript can be accepted after minor editing before publication.